# A two-dimensional Dirac fermion microscope

Peter Bøggild[1], José M. Caridad[1], Christoph Stampfer[2], Gaetano Calogero[1], Nick Rübner Papior[1] & Mads Brandbyge[1]

The electron microscope has been a powerful, highly versatile workhorse in the fields of material and surface science, micro and nanotechnology, biology and geology, for nearly 80 years. The advent of two-dimensional materials opens new possibilities for realizing an analogy to electron microscopy in the solid state. Here we provide a perspective view on how a two-dimensional (2D) Dirac fermion-based microscope can be realistically implemented and operated, using graphene as a vacuum chamber for ballistic electrons. We use semi-classical simulations to propose concrete architectures and design rules of 2D electron guns, deflectors, tunable lenses and various detectors. The simulations show how simple objects can be imaged with well-controlled and collimated in-plane beams consisting of relativistic charge carriers. Finally, we discuss the potential of such microscopes for investigating edges, terminations and defects, as well as interfaces, including external nanoscale structures such as adsorbed molecules, nanoparticles or quantum dots.

[1] CNG, DTU Nanotech, Department of Micro- and Nanotechnology, Technical University of Denmark, Ørsted Plads, Building 345C, Lyngby DK-2800, Denmark. [2] JARA-FIT and 2nd Institute of Physics, RWTH Aachen University, Aachen 52074, Germany. Correspondence and requests for materials should be addressed to P.B. (email: peter.boggild@nanotech.dtu.dk).

Graphene has proven to be an ideal host for spectacular mesoscopic effects, in particular after the introduction of hexagonal boron-nitride encapsulation[1–3] and efficient cleaning methods[4–7], which reduces scattering rates towards the theoretical limits as mainly defined by electron–phonon[2,5,8,9] and electron–electron[10,11] interactions. At low temperatures, encapsulated graphene with mean free paths of up to 28 μm has been reported[12], whereas the phase coherence length of several micrometres can be reached[13,14].

Efforts have mainly been devoted to explore the Dirac fermion counterpart of conventional mesoscopic phenomena, occurring in finite effective mass electronic systems at cryogenic temperatures: integer[9,15] and fractional[16] quantum Hall effect, weak localization[17,18], Fabry–Perot oscillations[13], commensurability oscillations[19–21], universal conductance fluctuations[22,23], Aharonov–Bohm oscillations[24,25], magnetic focusing[13,26,27] and quantized conductance[28,29]. Here, several factors set graphene apart from conventional two-dimensional (2D) systems. First, the linear gapless dispersion of graphene[30] gives rise to qualitatively different behaviours, such as Klein tunnelling[31,32] through potential barriers and Berry phase[9,15] in magnetic fields. Klein tunnelling allows Dirac fermions to penetrate high and wide barriers with zero reflection, which makes scattering at energy barriers resemble the transmission of light across an interface with an effective refractive index that can be tuned to negative values. Simply put, a single p–n junction constitutes an electron lens capable of focusing a beam of electrons, which was predicted for Dirac fermions by Veselago[33], shown to apply to graphene by Katsnelson et al.[31] and Cheianov et al.[32], and observed experimentally by numerous groups[6,13,27,34,35]. Second, in contrast to a 2D electron gas at the buried interface of GaAs and GaAlAs semiconductor crystals, graphene can be considered an open 2D electron gas, exhibiting highly tunable properties that strongly depend on its environment, as observed in Moiré superlattices[20] for graphene on hexagonal boron nitride and trigonal warping of the energy bands in bilayer graphene[36]. Third, momentum relaxation mean free paths may reach up to micrometre scale at room temperature[2,3,12,14,37], hinting that ballistic transport could lead to new opportunities for electronics[38,39] and optoelectronics[40,41]. It was recently shown that the electron–electron scattering length, $\ell_{ee}$, at elevated (room) temperature can be significantly smaller than the elastic mean free path, $\ell_{mfp}$, as well as typical device dimensions, $W$, for ultraclean samples. It has been pointed out that high-temperature transport resembles viscous flow and may be understood in terms of hydrodynamics, see for example, refs 10,42. Some microscopic manifestations of ballistic transport such as magnetic focusing[26] and negative differential resistance[43], however, have been shown to survive at room temperatures. In the following, we exclusively focus on cryogenic temperatures, where both electron–electron and electron–phonon scattering processes are strongly suppressed, and where an upper limit for the mean free path is yet to be determined. With the steadily improving graphene device quality and the numerous confirmations that graphene is capable of supporting transport in the mesoscopic regime ($\ell_{mfp} > L \gg \lambda_F, \lambda_\phi$)[2,6,12,19,26,34,35,44,45], we find that complex instruments that utilize relativistic charge carriers for practical purposes has become realistic.

An ordinary scanning electron microscope (SEM) is an extreme, yet familiar application of ballistic electron transport, which since its early incarnations more than 50 years ago has been a cornerstone of micro- and nanotechnology, surface and material science, as well as many branches within natural sciences. The electron microscope is based on four functions: emission, focusing, deflection and detection of a focused beam of ballistic electrons in a vacuum chamber, with the aim of analysing the shape, structure and chemistry of crystals, surfaces and small objects. The operation and individual components of an electron microscope in fact possess a striking number of similarities with state-of-the-art Dirac electron optics devices[6,19,26,34,46] and we note that the essential components and functions needed to realize such an instrument have been demonstrated experimentally. In particular, two essential electron optics components were very recently proposed: the absorptive pinhole collimator by Barnard et al.[47] and the parabolic lens by Liu et al.[48].

We examine here such an apparatus: a 2D electron microscope, based on the combination of elementary electron-optics components in a graphene device, which we in the following refer to as a Dirac fermion microscope (DFM). In this hypothetical intstrument, Dirac quasiparticles move in straight trajectories within 2D graphene rather than within a three-dimensional vacuum chamber. Electron beams may be focused onto intrinsic features such as grain boundaries, edges and defects, interfaces, contacts and edge terminations, and extrinsic nanoscale structures such as adsorbed molecules, nanoparticles, quantum dots and plasmonic superstructures to study their properties through their interaction with the electron system. Semiclassical ballistic calculations are effective in describing the overall magnetotransport characteristics in the mesoscopic limit[49] and are used to compare concrete architectures and designs suited for different types of target objects and applications.

## Results

**Anatomy of the DFM.** It is instructive to revisit the conventional SEM and its components (see Fig. 1a). In an electron gun, electrons are extracted from a metal by thermal or field emission, and collimated and accelerated by electrical fields and apertures. The electron beam is then focused into a small spot on the target surface by tunable electrostatic or magnetic lenses, which can be scanned across the surface by another set of magnetic or electrostatic deflectors. A detector located nearby captures either backscattered or secondary electrons returning from the irradiated surface, allowing an image to be generated.

In the following we consider how these four tasks may be carried out with a graphene device, using an arrangement such as illustrated in Fig. 1b. The 2D vacuum chamber is provided by graphene itself; at sufficiently low temperatures, the mean free path can be at least several tens of micrometres[2,12]. As the lithographic resolution offered by high-end electron beam lithography can be of order 0.01 μm, there are nearly four orders of magnitude difference between the dimensions of the components and the characteristic transport lengths, such as the Fermi wavelength, which is $\lambda_F \approx 2\pi^{1/2} n^{-1/2} \approx 35$ nm at a carrier density of $n = 10^{12}$ cm$^{-2}$, typical numbers for ballistic graphene devices. The 2D analogy of an electron gun can be realized by combining the functionality of basic components such as ballistic point contacts, apertures[47], Veselago lenses[32,34] and superlattice collimators[50,51]. Focusing of Dirac fermions can be carried by p–n junctions, where the carrier density can be controlled by electrostatic gates independently in the p- and n-doped regions. For the deflection of the beam, a perpendicular magnetic field provides a highly predictable means of controlling electron motion, as demonstrated in magnetic focusing[13,26,35,44,47,52] and snake states[6,45]. Detection can be done by large catch-all electrodes or by arrays of smaller electrodes to provide position or angular-resolved measurements[34]. Figure 1c shows a trajectory density plot (arbitrary scale) in the proposed DFM calculated by a semiclassical Monte Carlo simulator (see Methods). The trajectory density corresponds roughly to the current density, as the Dirac fermions have a constant velocity. The electron gun is

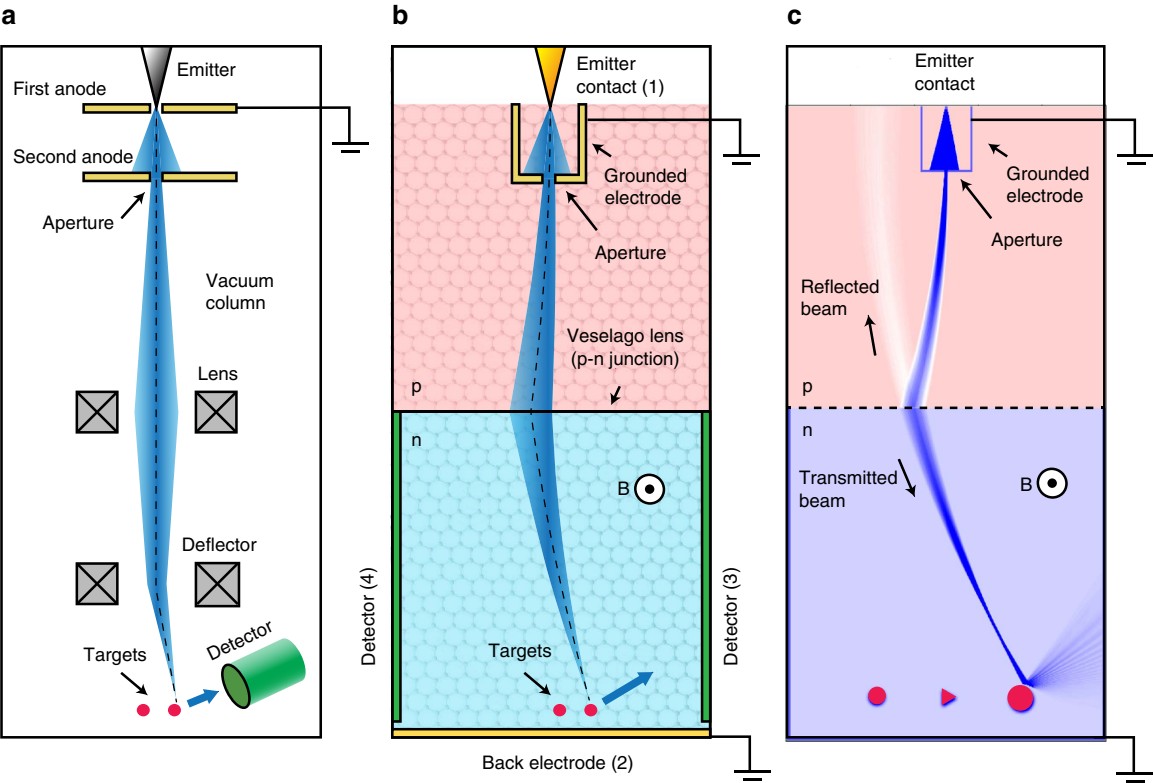

**Figure 1 | Conventional SEM versus DFM. (a)** Illustration of a SEM with the electrons travelling in straight lines inside a vacuum column, from the emitter, through an aperture and a set of lenses that focus the electrons into a sharp beam at the surface. A deflection lens directs the electrons to a specific spot on the surface, from which they scatter back and are measured by a detector. **(b)** The 2D equivalent is a graphene device, where a narrow metal contact or an opening plays the role of the electron emitter, while a p–n junction provides guiding/lensing of the electron trajectories. A back electrode (2) collects electrons that are not intercepted by a target. Backscattered electrons are picked up by side electrodes (3 and 4), allowing an image to be formed by measuring the transmission current. **(c)** Trajectory or current density from ballistic Monte Carlo simulation of a DFM, where carriers are injected from a point-like emitter contact (1), collimated by an aperture and focused by a symmetric p–n junction electron lens. Part of the beam is reflected from the p–n junction as indicated. The electron beam is seen to backscatter from one of the three targets, giving rise to a current flowing between electrodes 1 and 3, instead of 1 and 2.

here a point injection contact and a grounded aperture[47], and the lens is a symmetric, linear p–n junction.

**Graphene as a 2D vacuum chamber.** The vacuum chamber must provide an unobscured path for the electron beam, with a low presence of scatterers and with sidewalls of the microscope removed out of the beam path. Most graphene electron optics devices published in literature, such as Hall bar or van der Pauw geometries, have sidewalls within a distance of $\lambda_{\mathrm{mfp}}$ from the current pathways. In the ballistic limit, reflecting walls may add a non-trivial background to the conductance signal of interest, which may become more complex in coherent conditions (see, for example, ref. 53, Fig. 4), analogous to reverberation and standing acoustic waves in a poorly dampened room. This may lead to a conductance background in the vein of weak localization, conductance fluctuations, Fabry–Perot oscillations and Aharonov–Bohm oscillations. If the confining potential has strong geometric symmetries, the conductance fluctuations can exhibit pronounced regular features. We suggest that a graphene vacuum chamber should be designed either with semi-infinite sidewalls, that is, $L \gg \ell_{\mathrm{mfp}}, \ell_\phi$ such that the electron momentum and phase are fully randomized before the carriers return, or by diffusive walls, that is, with non-specular reflection to suppress unwanted ballistic and coherent reflections. Recently, electrically

grounded electrodes were shown to act as walls that remove carriers from the vacuum chamber and prevent them from returning to the main path of the beam[47].

Sidewalls are themselves relevant as objects of study with a DFM, given their importance for ballistic transport and devices depending thereof. Magnetic focusing of Dirac fermions has already been used to characterize the edge roughness and scattering properties of lithographically defined graphene edges[46,47]. An intriguing possibility is to probe pristine microcleaved edges; these offer near-zero structural disorder with either armchair or zig-zag structure that have been predicted to exhibit distinctly different scattering properties[54].

**Electron guns.** The electron gun is the component, or collection of components, which together generate a collimated, intense beam of electrons. In graphene devices, electrons are directly injected by metal–graphene contacts[6,12,35,52], or by ballistic graphene contacts where the metal–graphene contact region is located outside the main device area[34,46]. Although both these contact types are suitable, they result in point-like injection, as the task of focusing an electron beam to achieve a narrow diameter at the target area is greatly simplified if the electrons are injected from a point-like source, in analogy with light and electron optics. In analogy with classical wave mechanics, a point contact will

have a wide distribution of injection angles[47], with the special case of a rectangular, ballistic contact having a cos $\theta$ distribution[55]. Although size effects such as conductance quantization[28,29], Fabry–Perot-like interferences and sidewall roughness in the electron emitter may alter the angular distributions, we consider here the mesoscopic limit, where coherence and diffraction effects do not mask the overall behaviour[56]. For electron optics as for any type of optics, apertures provide straightforward means of reducing the angular spread of the beam, which was recently demonstrated experimentally by Barnard et al.[47]. The authors showed that a metal contact aperture connected to electrical ground can be used to obtain a significant reduction of the beam divergence in a graphene device, while at the same time removing stray electrons from the device.

**Electron lenses and focusing.** Klein tunnelling has been extensively described in literature, for example, refs 31–33, as well as comprehensively reviewed by Allain and Fuchs[57]. Here we outline just a few particularly relevant features. Klein tunnelling is the suppression of backscattering due to pseudospin conservation for an electron impinging on a potential step or barrier so that it will be transmitted with unity probability as illustrated in Fig. 2a. Although the transmission is unity for incident angle $\theta_i = 0$, the angular transmission function depends on the width $w$ of the potential step compared with the electron wavelength, translating into an effective smoothness $\alpha = k_F w$ of the step. For a hard, symmetric potential step, $k_i = k_r$ and $\alpha < 1$, the angular distribution[57] is given by $T(\theta_i) = \cos^2 \theta_i$. For a carrier density of $n = 10^{12}\,\mathrm{cm}^{-2}$, which we use throughout the simulations, the Fermi wavelength is $\lambda_F \approx 35\,\mathrm{nm}$, whereas the effective

smoothness $\alpha$ is unity for a width of $w = 5.6\,\mathrm{nm}$. Analytical expressions for the optical properties of potential steps and barriers in the sharp and soft limits are known for a number of situations[57], and we use here Cayssols[39,57] interpolation formula that connects the two regimes, see Fig. 2b. As illustrated in Fig. 2c, the incident and refracted angles, $\theta_i$ and $\theta_r$, follow the Snell–Descartes law:

$$k_i \sin \theta_i = -\,k_r \sin \theta_r \tag{1}$$

where $k_i$ and $k_r$ are the Fermi wavenumbers corresponding to the carrier densities in the two regions, $n_i$ and $n_r$, and the effective negative refractive index is given by $-(n_i/n_r)^{1/2}$. The transmission probability is unity for perpendicular incident angle, $\sin \theta_i = 0$, and the case of reflection gives $\theta_r = \pi - \theta_i$, as illustrated in Fig. 2c. Although p–n junctions constitute electron lenses capable of focusing a beam of electrons in graphene, steep and smooth p–n junctions behave very differently and should serve different purposes in the 2D electron microscope (see Fig. 1a). The steep p–n junction provides high transmissivity with a broad angular acceptance window and is well suited for electron lensing, where reflection is best kept at a minimum. A smooth p–n junction has a strong tendency to filter oblique angles, which can be used to collimate beams[51] as well as for electron guides and mirrors[52].

Such a p–n junction can be formed by electrostatic gating, such as the split bottom gates[6,58] or top gates[34], or by chemical gating, where metal islands deposited directly on top of the graphene provide the charge transfer necessary to induce potential steps or barriers[59]. For electrostatic gates, the width of the p–n junction can be controlled by the thickness of the dielectric spacer between the graphene and the gate[34], which suggests that an architecture

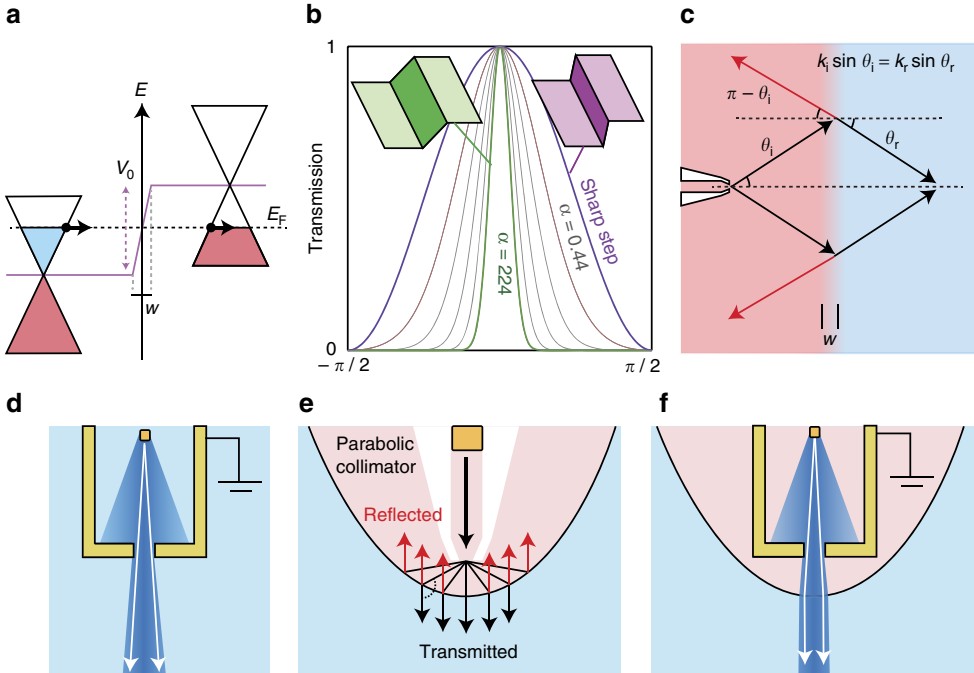

**Figure 2 | Electron gun and optics.** (**a**) Illustration of two Dirac cones offset by a potential step of height $V_O$. The width of the potential step is $w$. Carriers moving from the left (n-type) to the right (p-type) perpendicular to the step, but cannot be backscattered due to conservation of pseudospin. (**b**) The transmission through a potential step depends strongly on $\alpha = k_F w$, with large $w$ exhibiting a larger chance of reflection at oblique angles. The analytical angular distributions[57] from sharp to soft ($w = 40\,\mathrm{nm}$ at $n = 10^{12}\,\mathrm{cm}^{-2}$) p–n junctions, with intermediate curves calculated by Cayssols interpolation curve[77] for $\alpha = 0.44$–244 are shown. (**c**) Electrons impinging on a p–n junction at an angle $\theta_i$ with respect to the normal, are transmitted/refracted according to Snell–Descartes law, or specularly reflected. (**d**) An aperture limiting the angular distribution. (**e**) A p–n junction lens with a parabolic shape and the point contact positioned in the focus, will co-align the transmitted trajectories. (**f**) A combination of an aperture and a parabolic lens produces an electron beam with a long focal length.

that supports both reflective/filtering and transmission/lensing components could be realized with two top and bottom dielectric layers of different thickness.

Until now, little has been done to explore the possibilities and properties of curved p–n junctions for electron optics, perhaps due to the fact that the linear p-n junction itself is perfectly capable of focusing a Dirac electron beam[33]. In the language of optics, the linear Veselago lens suffers from significant aberration for non-zero magnetic fields, which makes it a less suitable starting point for a scanning beam microscope based on magnetic deflection. Inspired by parabolic mirrors that convert divergent rays from a source placed at the focus point into parallel rays, we employ the same principle to collimate and manipulate the electron beam as illustrated in Fig. 2e. A symmetric, parabolic p–n junction with the point contact placed in the focal point constitutes a parabolic electron gun with perfect collimation of transmitted electrons $\theta_r = 0$ for all $\theta_i$ according to equation (1) as well as reflection of trajectories with $\theta_r = -\pi$. The parabolic p–n junction greatly reduces the angular spread and increases the focal depth of the electron beam, ultimately offering a possibility of maintaining a parallel, focused beam across large scanning fields with much reduced aberration. Recent work by Richter and colleagues[48] solves the real-space Greens function in a tight-binding framework to analyse the detailed properties of the parabolic p–n junction. They predict that a parabolic p–n junction is capable of co-aligning a beam of Dirac fermions, and that such a beam can be deflected by a weak magnetic field without losing the collimation.

**Deflection.** A perpendicular magnetic field provides a means of deflecting the path of ballistic electrons in a way that is predictable and easy to control, as demonstrated for magnetic focusing in large number of experiments in both III–V and graphene mesoscopic systems[44,46,47,49,52,54]. The cyclotron radius is given by $R_c = \hbar k_F e^{-1} B^{-1}$. It follows from $k_F = (\pi n)^{1/2}$ that the cyclotron radius scales with $n^{1/2}$, whereas the cyclotron motion changes direction by reversal of either B or carrier polarity, that is, from n- to p-doped regions. In magnetic focusing, wide distributions of injected electrons lead to trajectories with pronounced caustics, which result in an oscillatory conductance with maxima at the point where the caustics intersect with the extraction point contact. In analogy with conventional optics, the width and shape of the caustic beam should indeed depend on the sidewalls and could therefore be used to extract information of the sidewall roughness[47] and electrostatic potential near edges[46]. As only certain discrete reflection points are being probed at each geometrically resonant magnetic field, imaging of the edge as such cannot be carried out, unless an array of contacts[34] is used to provide spatial or angular resolution. Although magnetic focusing is normally carried out at intermediate magnetic fields, $R_c \lesssim L$, where $L$ represents the characteristic dimensions of the sample, we consider in this work exclusively the low magnetic field limit, that is, $R_c \gg L$, where only slight bending of the electron beam is used to scan the focused beam spot across a feature; this situation closely mimics the image formation in a SEM. An example of how this could be done is shown in Supplementary Fig. 2.

**Detectors and image formation.** The simplest detector consists of a point contact. Imaging requires the extraction of the spatial distribution of some characteristic property, following an interaction between the sample and the probe, which in this case is a beam of Dirac fermions. Although it is tempting to use an array of electrodes to pick up spatial information, similar to the charge coupled device sensor in a transmission electron microscope, we consider a simpler scenario here that bear some resemblance to the architecture of a SEM, see Fig. 1. We find that two large catch-all contacts (left and right) are sufficient to generate images that discern between sizes, shapes and orientations of obstacles with sizes comparable to or above the beam diameter. The back-plane is an absorptive electrode, which can be used to catch the carriers similar to a Faraday cup, allowing the beam current to be measured as in a conventional SEM. The images are constructed from the transmission between source and detector/drain electrodes as a function of magnetic field, which is the number of electrons arriving at the detector electrode, divided by the number of electrons emitted by the source. The transmission includes the ballistic contact resistance of the electron gun, which depends greatly on the exact gun configuration. However, as in a conventional microscope, only the relative intensity variations matter; hence, contrast and brightness adjustments are needed to achieve a useful image. Transmission values and trajectory (current) densities are therefore arbitrary scale.

**Scanning DFM microscopy with single emitter.** We have performed extensive Monte Carlo trajectory simulations to compare different microscope configurations and have analysed the behaviour of the microscope components and simulated the image formation in scanning DFM (see Methods for details). The lens components we combine, to obtain control of the shape, angle, divergence and position of the beam, are the grounded aperture (absorptive pinhole collimator)[47], the parabolic p–n junction for co-aligning the electron trajectories and the linear Veselago lens[32] for refocusing a divergent beam. The beam profile and angular distributions, as well as linearity of beam position with respect to magnetic field are shown in Supplementary Fig. 1.

In Fig. 3, we compare simulated images of the three microscope configurations. The target consists of three antidots; two circular shapes of different diameter and a triangular shape. Figure 3a–c show the overlaid trajectory density as well as the transmission between source 1 and either drain 2 (the backplane electrode) or drains 3 or 4 (detector electrodes), with two magnetic fields for each microscope configuration. Owing to the constant carrier velocity, $v_F \approx 10^6\,\mathrm{m\,s^{-1}}$, the trajectory density is directly proportional to the semiclassical (non-coherent) current density. Figure 3d,e show the transmission $T_{12}$, $T_{13}$ and $T_{14}$ as a function of the magnetic B-field, between injection electrode 1 and the three drain electrodes 2, 3 and 4, respectively. As the magnetic field sweeps the electron beam across the target, shape-specific features appear in the magneto-transmissions $T_{13}$ and $T_{14}$, which in the following will be referred to as transmission images. The spherical scatterers generate two peaks of equal height and shape, spaced roughly by the projected width of the circular potential. The triangle gives rise to a narrow peak from the leftmost corner and a broader peak or plateau corresponding to the flat, right-hand side. Although these features are recognizable in all three configurations, they are more sharply defined for Fig. 3b compared with Fig. 3a, which is distorted by the broader angular distribution of incoming electrons, and Fig. 3c, which is characterized by a long focal length, but also a wider beam diameter. The relative transmission for Fig. 3b and to a lesser extent Fig. 3c is decreasing with magnetic field, as large field magnitudes cause the electron trajectories to intersect the p–n junction at higher angles, thereby reducing the transmission probability. The Supplementary Note 4 shows the imaging process for two different aperture sizes, with only minor difference in image quality.

In general terms, a symmetric Veselago lens creates a mirror image of the point-like source, which is distorted by varying the

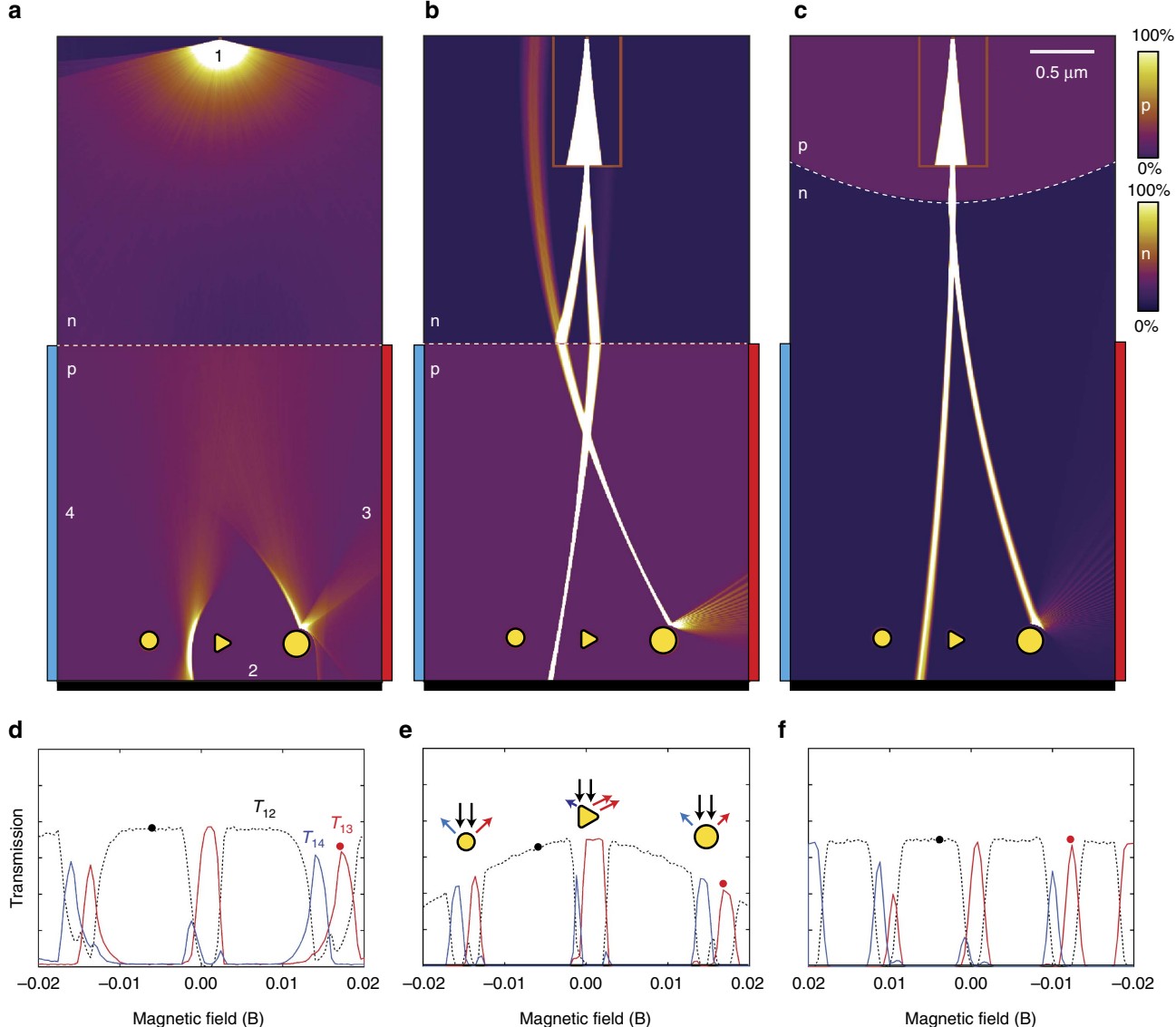

**Figure 3 | Imaging with Dirac fermions.** Current density maps of three DFM microscope configurations, (**a**) Veselago, (**b**) Pinhole/Veselago and (**c**) Pinhole/Parabolic lens. In each case, the current density for two different magnetic fields are overlaid, indicated with black and red dots in panels below. Colour scales ranging from 0% (zero current density) to 100% (high current density) are used to visualize zones with different carrier polarity p and n. (**d–f**) The dashed black curves show the transmissions from electrode 1 to the back electrode 2, whereas red and blue curves represent carriers exiting at the right (3) and left (4) electrodes, respectively, as also indicated in **a**. The magnetic field axis is intentionally reversed to correctly align the red and black dots in **f** with the simulated electron beams in **c**. The transmission images from the three target objects are distinctly different, with the spherical objects leading to relatively symmetric peaks and the triangular object giving rise to a sharp transmission peak from reflection at the left corner of the triangle, and a broader peak or plateau corresponding to reflection from the flat, right-hand side of the triangle. Of the three selected configurations, the configuration a exhibits the largest aberration and image distortions, configuration b the sharpest features due to highly focused beam, whereas c maintains a constant transmission current across the image field. The three configurations are analysed with respect to beam profile, angular distribution and linearity with respect to the magnetic field in Supplementary Note 1 and Supplementary Fig. 1. The imaging is showing during operation in Supplementary Movie 1.

carrier density on one side[32], or by a finite magnetic field. Owing to the pronounced caustics resulting from Dirac fermions passing a linear Veselago lens, it is still possible to generate an image with a high spatial resolution. Moreover, the caustic beam spot position is nearly perfectly linear in magnetic field, see Supplementary Fig. 1. This situation can be augmented by introducing other lens components. An aperture will limit the angular spread by blocking most of the diverging beam, whereas a parabolic lens reduces the angular spread by co-aligning the beams. The combination can give a beam with exceedingly low angular spread and therefore very long focal depth.

Another strategy to provide spatial image resolution is to use arrays of collector electrodes. This approach could be useful for imaging by magnetic focusing, where the reflection point can be swept across an edge, whereas the collector array picks up the backscattered electrons. An example of such a magnetic focusing-based imaging of edge roughness using five electrodes is shown in Supplementary Fig. 2 and Supplementary Movie 5.

**Imaging of Veselago dots.** A different type of scattering potential is the closed p–n junction, which we here term a Veselago dot (VD). Gutierrez *et al.*[60] found that few-nanometre graphene p–n junctions formed in continuous graphene on copper due to local variations in surface interactions and surface potential showed the signatures of distinct quantum states corresponding to periodic

polygonal trajectories inside the potential. Caridad et al.[59] formed arrays of VD by depositing metal dots directly on graphene to locally pin the electrostatic potential, leading to sharp, circular p–n junctions by tuning of the back gate to the opposite polarity. The VD-like behaviour was corroborated by measurement of Mie-like scattering of the electron waves on tilted arrays of VD with respect to the current direction. VD are lens-like potentials that can reflect, trap and re-emit electrons depending on their incident angle, width and height of energy barrier and serves a double role as interesting objects of study, as well as potentially useful and simple electron optics components with naturally hard p–n junctions and no need for a gate dielectric.

Figure 4 shows simulated current density plots for three different spherical potentials: p–n junctions with $w = 2.5$ nm and $w = 40$ nm, as well as a fully reflective disc-shaped potential. At certain magnetic fields, the focused beam leads to pronounced,

internal scattering patterns and narrow emission jets at the reflection points, which can be thought of as classical counterparts of jets in optical cavities[61]. For glancing incident angles, the $w = 40$ nm VD reflects all trajectories, whereas for the $w = 2.5$ nm VD, high-order polynomial closed trajectories appear inside the boundary. The caustic emission jets lead to local transmission maxima (red dots) at the electrode opposing the main reflection direction (blue dots). These distinct signatures of the scattering profile depends strongly on the width $w$, shape and height of VD, and can be used to analyse the properties of p–n junctions down to very small widths[59]. Supplementary Movie 2 shows the development of current density corresponding to Fig. 4.

**Scanning DFM microscopy with multiple emitters.** For a coherent electron system, such as high-quality graphene at

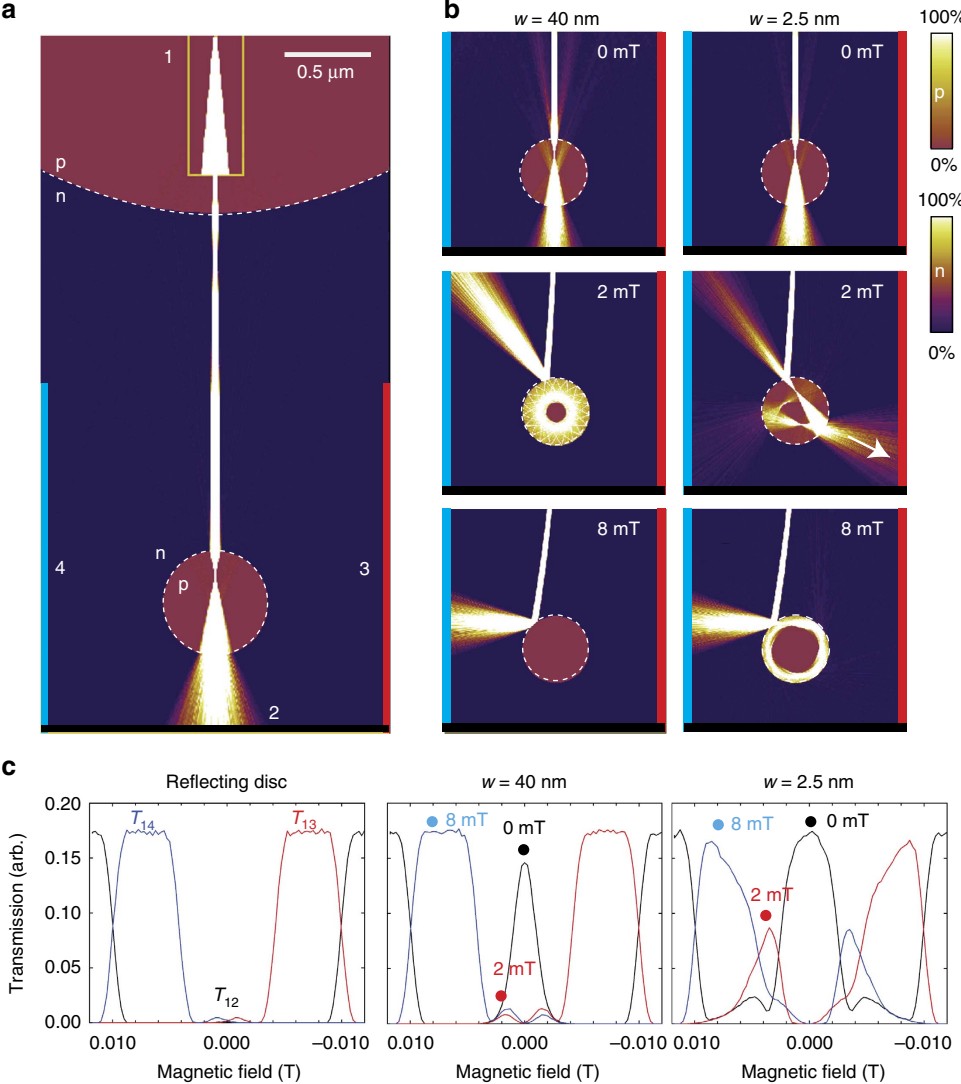

**Figure 4 | Imaging of VDs.** (a) Focused electron beam directed towards a circular p–n junction, a so-called VD, using the DFM configuration in Fig. 3c. The transmission into the VD depends on the angle of the incoming beam with respect to the junction wall. At zero incidence angle (along the centre line), the beam is nearly fully transmitted through the dot, before hitting the back electrode (2, black). (b) shows the current density for two different p–n junction width, $w = 2.5$ nm (close to hard-wall limit) and $w = 40$ nm. Hexagonal boron nitride layers thickness is typically in the range 10–100 nm[34,47]. Colour scales ranging from 0% (zero current density) to 100% (high current density) are used to visualize zones (p,n) with different carrier polarity. (c) The calculated transmission coefficients, $T_{12}$ (black curve), $T_{13}$ (red curve) and $T_{14}$ (blue curve), for the three drain electrodes as a function of magnetic field, plotted for two VD with $w = 40$ nm and $w = 2.5$ nm, as well as a reflecting disc. The magnetic field values of 0, 2 mT and 8 mT are marked as black, red and blue dots. At 2 mT, the current jet emitted towards electrode 3 for the $w = 2.5$ nm VD is shown as a large peak (red curve) in the transmission image. At $w = 40$ nm the jet is much weaker.

cryogenic temperatures[22,62,63], classical periodic orbits are directly related to bound quantum states and can be used to predict the energy level spectrum[64] and transport properties[65,66]. For an open or semi-open quantum billiard[62]—a constant confinement potential with hard or soft sidewalls—electrons are injected via point contacts at the edge of the potential, which makes stable periodic orbits classically inaccessible without scattering[66]. The injection point contact itself will scatter any polygonal orbit on the first roundtrip. For the VD, the situation is entirely different; the semitransparent p–n junction will indeed allow carriers to be injected directly into a periodic orbit, as shown in Fig. 4.

In Fig. 5a, we introduce a microscope configuration that mimics the nearly parallel electron beam of a conventional SEM, allowing the beam to pass through the target area without changing focus or angle. The aperture is placed at the focus of the parabolic lens and the single emitter is replaced by an array of $N$ point emitters. The beam will be aligned parallel to the centre axis by passage across the parabolic p–n junction and the slight divergent beam can optionally be refocused using a symmetric Veselago lens, as in Fig. 3a. The result is a narrow beam that can be moved in coarse steps between $N$ positions, corresponding to the $N$ emitter electrodes. The beam position can be further fine-tuned by the magnetic field, leading to seamless coverage of the image plane, as indicated with the dashed line. Full-range scanning requires stitching of the regions defined by the selected emitter electrodes by fine-tuning of the magnetic field. Together, the emitter array, the aperture and the parabolic lens constitute a composite electron optics electron gun with a low aberration, near-parallel beam and a large scan range.

Figure 5b depicts the transmission image profile of circular and triangular reflective targets similar to those imaged in Fig. 3, with the $T_{12}$, $T_{13}$ and $T_{14}$ curves shown both before (dashed lines) and after (full lines) compensation for the unavoidable current decrease at large offsets, where the collimated beam intersects the parabolic lens at oblique angles.

The lower panel shows the position of the beam at the electrode depending on emitter electrode and magnetic field between $ca.\ -1$ and 1 mT, with the white marker indicating the sequence of emitter position (1–9) and B-field used to achieve a continuous scan. This strategy is similar to the separate deflector systems for coarse and fine alignment in an electron beam lithography system, where small regions are stitched together into large continuous regions, to avoid large beam deflections and image distortions. After correction, the transmission image of the circular potential is symmetric, whereas the image of the triangular potential shows a pronounced flat part (red curve), corresponding to the flat part of the triangle, facing right.

Figure 5c shows a wide-angle distribution of trajectories being co-aligned by a parabolic lens and directed at a circular p–n junction with $w = 2.5$ nm. A clear pattern of caustics is visible, which agrees exactly with those predicted by differential geometry (see Supplementary Note 2 and Supplementary Fig. 3). In Fig. 5d,e, a focused beam is directed at the injection points for the triangular and square polygonal closed orbits[60], leading to strong caustic resonances and collimated emission jets at the corners.

Figure 6 shows transmission images of 200 nm-wide VD structures, with effective width of the p-n junction ranging from $w = 2.5$ nm to $w = 40$ nm (see Fig. 2b). Figure 6a shows current density for four selected magnetic fields, indicated with arrows on the transmission image (Fig. 5b). Although the forward transmission peak in the coefficient $T_{12}$ (black curve) due to Mie-like scattering[59,67] is relatively insensitive to $w$, $T_{13}$ (red curve) and $T_{14}$ (blue curve) show distinct features related to the intensity of the caustic jets, with a clear dependence on $w$. This shows that the spatial mapping can indeed provide detailed

information of the nanoscopic properties of the scattering potentials.

In Supplementary Note 3 we discuss how a DFM may be fabricated using van der Waals assembly[37]. In Supplementary Fig. 5 we show that the transmission images are robust towards increasing the aperture size, which can be advantageous to reduce diffraction effects for low carrier densities, large Fermi wavelength and narrow apertures[47]. Supplementary Movie 3 illustrates current density variations during imaging of two different VDs and a reflecting disc, and Supplementary Movie 4 shows imaging and caustic jets of a large VD using multiple emitter Dirac fermion microscopy.

## Discussion

We have described a class of devices that through control and scanning of a beam of relativistic carriers allows a form of in-plane scanning microscopy in two dimensions, and show examples of how imaging of different types of objects could be carried out and how different beam profiles and behaviour can result from the interplay of the electron optics components. Obviously, these can be combined in far more ways than shown here.

There are several issues that should be addressed. First, the question of whether the graphene can provide the disorder free vacuum chamber at low temperatures. As of now, the highest reported elastic mean free path in graphene is 28 μm[12]; however, so far no fundamental upper limit has been reported. It appears that the cleanliness of the interfaces, quality of the materials and strain inhomogeneities are the limiting factors, as the intrinsic electron–phonon scattering processes freeze out at cryogenic temperatures. For III–V heterostructures, mean free paths in excess of 100 μm have been reported[68]. The mean free paths in cryogenic graphene could be even longer.

One potential problem with narrow beams that are guided across long distances by small magnetic fields, is that the sensitivity to weak disorder potentials and variations in the carrier density could lead to signal noise and beam broadening effects. Even with ballistic mean free paths extending beyond the sample size, the beam may still follow irregular paths, as reported in III–V heterostructure 2D electron gas by Jura et al.[69]. Second, it is pertinent to consider whether the state of the art of device fabrication can deliver the precision and reliability to make actual electron optics instruments feasible. Although strain inhomogeneities and interfacial contamination present some of the more challenging issues for van der Waals heterostructure assembly[37], the field is developing rapidly. The key techniques for high-quality van der Waals heterostructure assembly[1,2,37,70], edge contacts[2] and patterned layers[37,71] were introduced and developed just a few years ago, and there is a significant effort in developing methods for scaling up the processes that presently rely on exfoliated materials. A suggestion for a possible fabrication scheme based on published techniques can be found in Supplementary Note 4.

Third, we consider the question regarding how quantum coherence will influence the operation of the microscope, beyond setting a limit for the image resolution in the tens of nanometre through diffraction effects. Coherence will for instance influence the angular distribution of electron beams passing through narrow apertures[47], leading to a broader beam according to the Huygens principle[34]. This can, however, be countered by increasing of the aperture size and the carrier density to reduce the Fermi wavelength. Our semiclassical simulations indicate that the image formation is robust towards increasing the aperture size, as we show in Supplementary Fig. 6. As pointed out above, our simulations represent the mesoscopic limit, where electron currents are well approximated by classical trajectories/ray

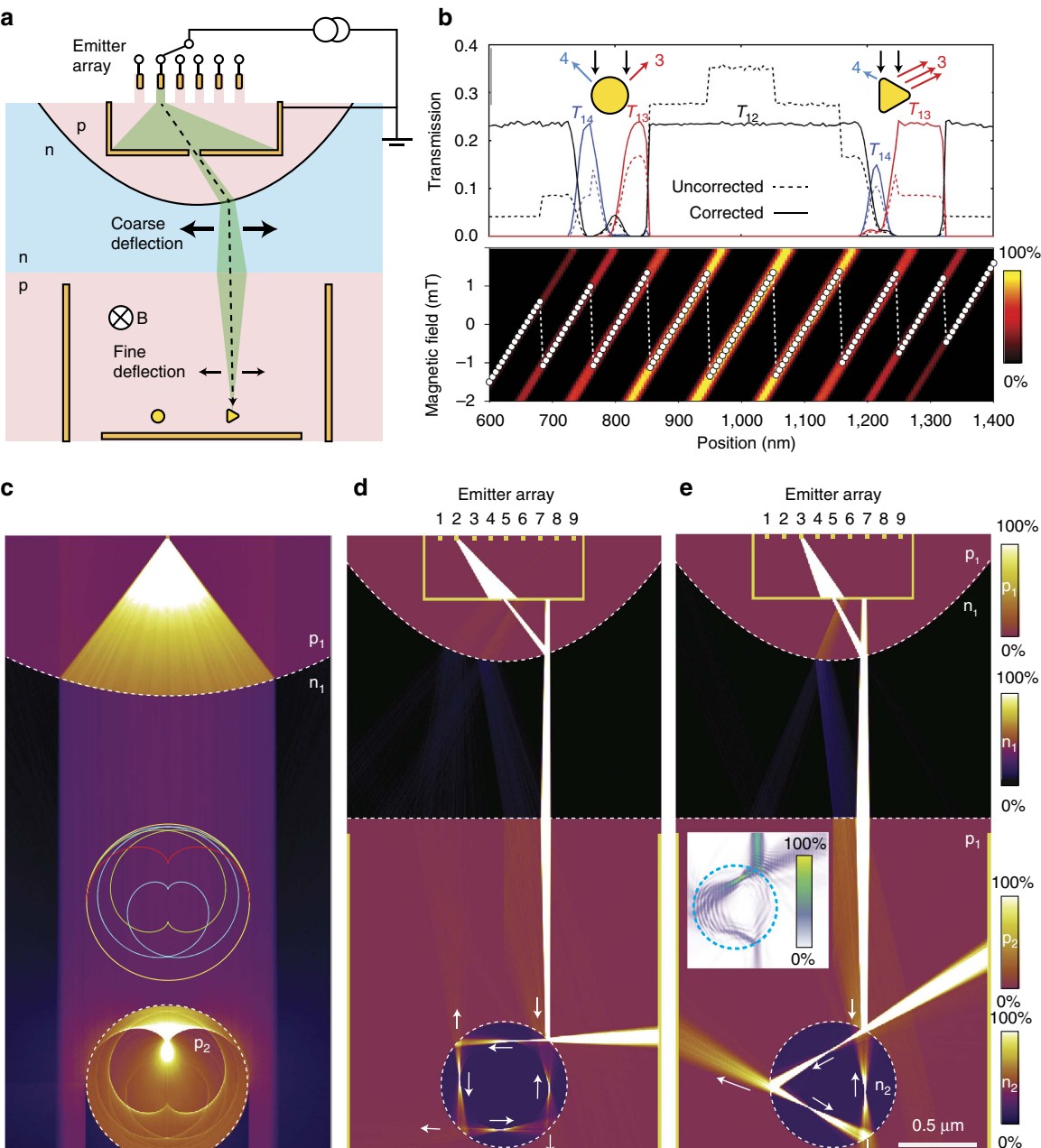

**Figure 5 | DFM with multiple emitters and coaligned beam.** (**a**) Illustration of the coaligned scanning DFM, where an array of emission electrodes and a parabolic p–n junction allows for lateral displacement of a vertically aligned beam of electrons. A small magnetic field provides fine adjustment of the beam position. (**b**) Transmission from emitter to electrode 2 (bottom), electrode 3 (right) and electrode 4 (left) before (dashed) and after (full lines) correction for emission current variations of the nine emitters. The lower panel shows the switching between emitter electrode (9,8,7,…,1) and magnetic field (y axis) needed to produce a continuous coaligned scan. (**c**) Parallel beam of electrons scattering on a circular p–n junction ($w = 2.5$ nm), producing the well-known caustic pattern of trajectories in a circular potential (see Supplementary Note 2). (**d,e**) Injection of current directly into square and triangular closed orbits. The carriers are transmitted out at the three corners, producing well-collimated jets. Colour scales ranging from 0% (zero current density) to 100% (high current density) are used to visualize zones ($p_1$, $n_1$, $p_2$, $n_2$) with different carrier polarity in **c**-**e**. The inset in **e** shows the bond current results from an atomistic transport calculation of a graphene VD with a similar ratio between diameter and $\lambda_F$ as used in the semiclassical simulation, resulting in a current density resembling the triangular closed orbit and a current jet emission pattern qualitatively in agreement with the semiclassical simulation. The quantum transport calculations are detailed in Supplementary Note 3.

tracing[35,56]. Upscaling of the lateral dimensions as a route to diminish diffraction and coherence effects will benefit directly from continued development of sample quality[2,12]. One exception is the scattering of Dirac fermion beams on small objects, where quantum coherence effects may cause significant deviations from the transmission signatures found from our semiclassical trajectory simulations. In Supplementary Note 3 we show maps

of the atomistic bond current from atomistic tight-binding calculations, for collimated Dirac fermion beams generated by parabolic lenses and narrow apertures. Three cases are presented, to illustrate the main concepts of this work: a caustic pattern corresponding to Fig. 5c, a triangular closed orbit in a circular p–n junction with jet currents corresponding to Fig. 5e and scanning of a focused beam using the magnetic field across a

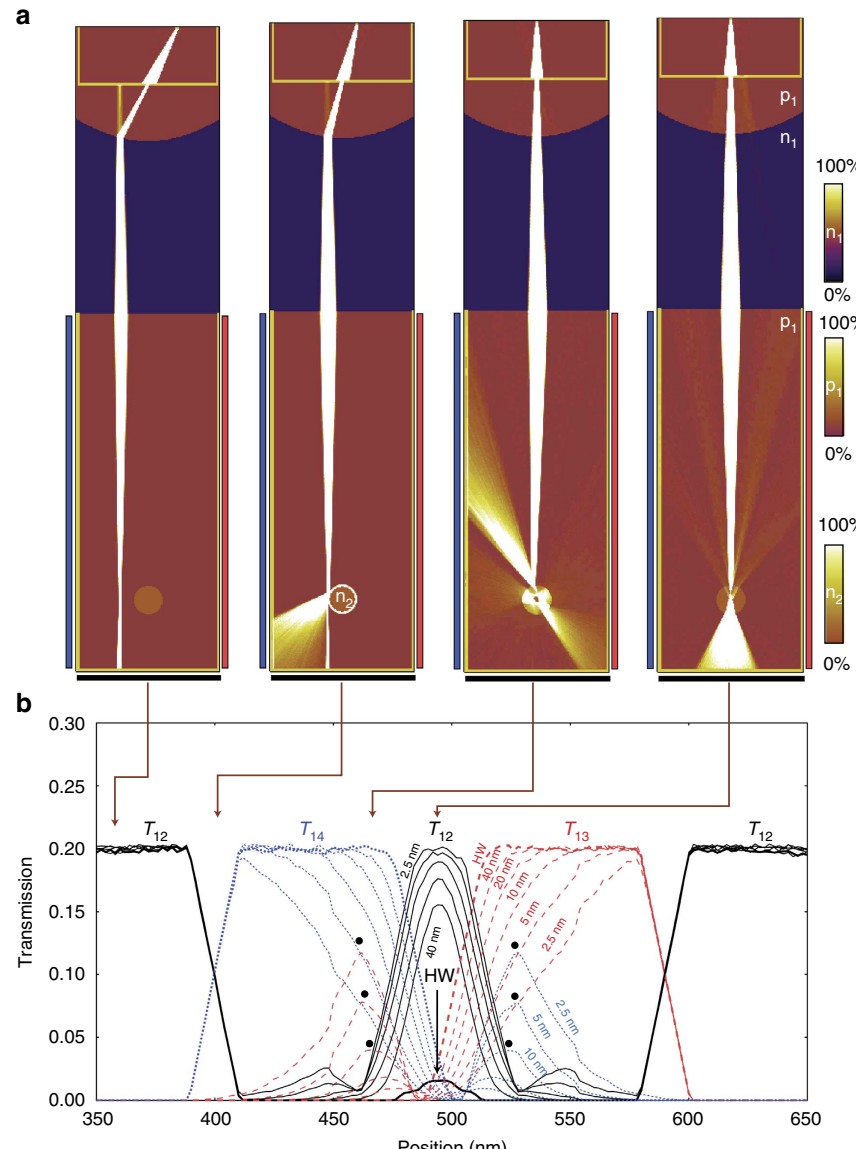

**Figure 6 | Imaging of VDs with different smoothness. (a)** Current density plots in a $1 \times 4.5\,\mu m$ DFM with 200 nm diameter VD targets for four different magnetic fields, indicated with arrows in the panel b below. The width of the p–n junctions is $w = 10\,nm$. **(b)** Development of transmission image for VDs with different width ($w = 2.5, 5, 10, 20$ and $40\,nm$), as well as hard-walled, reflective potential (HW), for the four magnetic fields simulated in **a**. The dotted, blue curves correspond to the transmission $T_{14}$, for electrode 4, whereas the dashed, red curves represent the transmission $T_{13}$, for electrode 3. The black, full curves show the transmission $T_{12}$ between electrode 1 and the backplane electrode 2. Although the zero-incident angle peak (at 500 nm) does not change significantly with $w$, the transmission side-bands caused by caustic jets, at 400–500 nm and 500–600 nm, changes dramatically from $w = 2.5\,nm$ to $w = 40\,nm$ (marked with black dots).

small circular VD, as in Figs 4 and 6. As expected, the simulations show that current density of structures which are large compared to the Fermi wavelength, reasonably resembles the classical calculations, but also that quantum coherence leads to bond current patterns with a richer emission and reflection structure, which may be utilized to extract more detailed information of targets than possible with semiclassical calculations.

In that sense, we anticipate that quantum coherence is indeed an opportunity for developing more advanced functionality of the DFM. With phase-coherent beams, interferometric and even holographic microscopy could give new insight in conductance fluctuations and weak localization, as now individual or sets of trajectories can be selected without the need for permanent wires, instead of ensemble averaged.

Along the same lines, we envision utilizing spin and valley degrees of freedom of graphene's charge carriers. For example, the long spin-life times in graphene[72,73] may enable the use of spin-polarized electron beams to study magnetic edge terminations, magnetic molecules or local proximity-induced spin–orbit interaction. For this, ferromagnetic contacts can provide for spin-polarized injection and detection. A step further one could also imagine to make use of point contact-based spin and valley filters[74] for increasing the functionality of the DFM. An interesting target for a spin polarized DFM would be strained nanobubbles[75] that exhibit huge pseudomagnetic fields and can act as spin filters and beam splitters[76]. Focused beams could possibly also be used to investigate interactions between layers in more complex, multilayered heterostructures[71].

Although the image resolution of a DFM can never rival those of established electron or scanning probe microscopies, the DFM could provide new insight in the details of microscopic scattering processes[67] and interactions with the environment, disorder, adsorbed molecules, quantum dots, which are crucial for sensing, electrons and optoelectronics applications.

In a broader perspective, the DFM embodies a wireless electron transport measurement system, where carriers can be injected, directed and focused onto objects of interest to perform a form of transport measurements without edge scattering, inflexibility and limitations of a permanent, hardwired physical wires.

## Methods

**Semiclassical and quantum calculations.** The Monte Carlo simulations use a variable time step Verlet numerical integration algorithm, with optional disorder and possibility of mixing numerical potential energy maps with coordinate-based, analytical geometrical boundaries to generate complex scattering landscapes. The target objects in Figs 3 and 5b are modelled as nearly hard-walled potentials obtained by convoluting a step potential with a Gaussian function of $<10\,nm$. In the simulation, the point contacts emit trajectories with uniform angular distribution (corresponding to a metallic contact), however, to optimize the calculation speed in certain configurations the distribution is artificially narrowed before passage through an aperture, see, for example, Fig. 5d. This only affects the computation time, as less time is spent on trajectories that will anyway hit the absorptive walls of the aperture enclosure. The transmissions between the electrodes are calculated by dividing the number of exiting trajectories with the number of emitted trajectories; we do not take into account the variations in ballistic contact resistance for the different electron gun configurations, as the relative change in transmission with position or magnetic field is sufficient for image generation. The p–n junctions are modelled using the Cayssol approximation formula[77] with width $w = 10\,nm$, unless stated otherwise, with the carrier density kept fixed at $10^{12}\,cm^{-2}$, for both p- and n-doped regions; these are typical values for electrostatically gated graphene. Although the simulation was carried out in $4\,\mu m \times 2\,\mu m$ or $4.5\,\mu m \times 1\,\mu m$ area in all simulations, the dimensions of our proposed devices can immediately be scaled up. Increasing all dimensions by a factor of $S$ will yield identical results by corresponding scaling of the cyclotron radius, that is, by changing the magnetic field or the carrier density by a factor of $S^{-1}$ or $S^{1/2}$, respectively. In a practical device, scaling the system up will reduce issues with diffraction, that is, through narrow apertures[47], but makes higher demands with respect to the mean free path and presence of small angle-scattering effects[69]; for fabrication of a real device, this is an important trade-off and a strong motivation to push device quality in a similar manner as for III–V 2D electron systems[68]. Atomistic tight-binding calculations were performed to examine the scattering of focused, collimated Dirac fermion beams on circular p–n junctions and provide a basis for comparison with the semiclassical calculations. The methodology and results of these calculations are described in Supplementary Note 3 and Supplementary Fig. 4.

**Data availability.** All relevant data as well as the computer code for the semiclassical Monte Carlo simulations are available from the authors.

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

## Acknowledgements

This work was supported by the Danish National Research Foundation (DNRF) Center for Nanostructured Graphene (DNRF103), the EU Graphene Flagship (604391) and DFF-EDGE (4184-00030). We are thankful for discussions with and suggestions from Peter Uhd-Jepsen, Antti-Pekka Jauho, Bjarke Sørensen Jessen, Lene Gammelgaard and Timothy J. Booth.

## Author contributions

P.B. conceived the DFM concept and carried out the semiclassical calculations, with significant contributions from the coauthors. G.C. performed the quantum transport calculations with M.B. and N.P. The manuscript was written by P.B. with assistance from the coauthors. J.M.C. contributed the caustic analysis. All authors discussed the results and commented on the manuscript.

## Additional information

**Competing interests:** The authors declare no competing financial interests.

