## [Peer Review File · Nature Communications]

Reviewers' Comments:

Reviewer #1:

Remarks to the Author:

This paper presents a novel microscopy technique based on simulations only. This is not serious. Microscopy is an experimental discipline with many Nobel prizes that were awarded based on experimental evidence. I cannot believe about a new microscopy technique until I see experimental results that are superior to existing techniques, like STM and electron microscopy. This paper should not be published in Nature.

Reviewer #2:

Remarks to the Author:

The authors discuss different schemes for realizing an electron microscope based on graphene Dirac electrons.

While the paper surely deserves publication in some form -- after some revision and proofreading, see below -- to my mind it is clearly not suitable for publication in Nat. Commun. The manuscript does not bring up any substantial new idea, it mainly exploits two already recently proposed ideas for creating Dirac electron beams -- Barnard et al. absorptive pinhole collimator (Ref. [45]), Liu et al. parabolic collimator proposal (Ref. [58]) -- to discuss how the resultant beam can be used for microscopy purposes in specific geometries. The collimator studied by the authors, see Figs. 2 - 6, is a simple "series combination" of the pinhole of [45] and the parabola of [58], compare with Fig. 1 from [45] and Fig. 1 from [58]. Though this is evident, it is not very clearly stated, and I find the presentation a bit misleading. Furthermore, the simulations are classical and hence much cruder than those of [58], which appear to be quantum mechanical. The analysis carried out in the experimental work [45] also appears more accurate than the present one, as diffraction effects are explicitly considered in [45].

A few additional remarks:

1. The beam bending in Fig. 1 (b), (c) is wrong, given the direction of the magnetic field. Checking all other figs. for similar problems is required.
2. The colour code of the images is never discussed. Even if arbitrary units are used, I think this should be done to improve clarity. For example in Fig. 3 "bright" should mean high ray (electron current) density, but this is also used for highlighting the targets. Furthermore, certain figure panels seem not to be referred to in the main text, e.g. Fig. 2 d: as far as I understand, in line 23 of p.3 the authors refer to 2 d, but actually mean 2 e. See point 1. about checking for similar mistakes.
3. Since graphene is not a topological insulator, what is meant with "topologically protected backscattering" in the introduction?
4. A scale on figs. such as 3 and 4 would be helpful, e.g. showing explicitly the Fermi wavelength in comparison to the size of the targets.
5. Some proofreading would help. A few examples: P. 7, line 19: "singular point"  "single point"; p.26, line 18: "absorptive"  "absorbptive"; p.27, line 4 "micron micron"  "micron" ...

Reviewer #3:

Remarks to the Author:

In this manuscript, the authors theoretically design a two-dimensional Dirac fermion microscope in graphene and simulate the detection and the image formation of objects by a beam of ballistic electrons in vacuum chamber. The design and the performance of the key components, including collimators, focusing lenses, deflectors and detectors, are analyzed in detail. The 2D Dirac fermion microscope proposed in this work is scientifically significant and promising for practical applications. However, the authors need to clarify the following issues.

1. We expect that objects to be studied and Dirac fermion microscopes (DFMs) don't necessarily locate in the same graphene platform. Just like a traditional optical/electron microscope, objects can be studied only when they are placed below object lenses. In the current simulations in the manuscript, I understand, the objects studied locate in the same graphene platform as the DFMs. I want to know whether the DFMs can work when objects locate in another graphene platform.
2. Is the spatial resolution of the DFMs comparable to traditional electron microscopes?
3. As a collimator, the aperture size in a metal contact aperture connected to electrical ground must be small in order to guarantee the high degree of collimation. Is the ratio of the electrons passing through the aperture to that emitted by the contacts enough high to make sure that the electron beam arriving at objects is strong?
4. Besides a parabolic lens, an aperture is first used to collimate electron beams. Whether do parabolic lenses work well only when the angular spread of incident electron beams is small? Please clarify this point in the manuscript.
5. Does "hBN" in page 2 refer to hexagonal boron nitride?
6. Some minor changes must be done in the following sentences:
 - 1) "where the metal-graphene contact region from the main device area" in page 7.
 - 2) "For 10^{12} cm^{-2} " should be explicitly expressed as "For the carrier density $n=10^{12} \text{ cm}^{-2}$ ".
 - 3) "as illustrated in Fig 2d" should be written as "as illustrated in Fig. 2e" in page 10.
 - 4) What do "according to (0.1)" refers to in Page 11?
 - 5) I can't find Fig. 1f and Fig. 3g that are mentioned in page 12 and page 14, respectively.
 - 6) "get a information" in page 15.
 - 7) (a-d) should be written as (a-c), in the caption part of Fig. 4.

Reviewer #1 (Remarks to the Author):

REV1: This paper present a novel microscopy techniques based on simulations only. This is not serious. Microscopy is an experimental discipline with many Nobel prizes that were awarded based on experimental evidences. I cannot believe about a new microscopy technique until I see experimental results that are superior to existing techniques, like STM and electron microscopy. This paper should not be published in Nature.

Not only do we respect and appreciate reviewer 1's opinion, we are also to some extent inclined to agree. The best way to present a novel idea for an instrument, is to show a working prototype, to show real experimental results, and demonstrate superior performance. We do hope that the reviewer will consider the following reasons why we believe that the situation is a bit different here, and that the work could indeed impact the field in a way that will lead to an interesting development of genuinely useful apparatus and scientific instruments, embedded in and supported by a very unusual technological platform, namely a two-dimensional solid-state material.

Need for experimental validation. One reason that we think it worth considering our work for publication in a highly visible, cross-disciplinary journal such as Nature Communications, is that essentially most of the optical components and all physics needed to realise the hypothesized DFM instrument, have already been demonstrated experimentally. We propose to construct a new type of scientific instrument, a first (to our knowledge) concrete apparatus or instrument employing ballistic electron optics, even though *graphene electron optics* has been a hot topic for more than a decade^{1,2}.

We have changed the formulation of a sentence in the introduction to better clarify this:

“The operation and individual components of an electron microscope (see Fig 1a) in fact possess a striking number of similarities with state-of-the-art Dirac electron optics devices³⁻⁷, and we note that the essential components and functions needed to realise such an instrument have been demonstrated experimentally.” (P4, L1)

Comparison with existing microscopies. Electron optics has been highly rewarding and successful area for fundamental research, a vehicle to explore the wonders of Dirac electronics and photonics, but we would like to urge our colleagues to push towards scientific instrumentation, where little or nothing has yet been done. The “open” nature of the graphene “vacuum chamber” is the key: the fact that direct interaction with adatoms, quantum dots, molecules and other objects can be induced by simply laying these on top. This gives opportunities and challenges that are very different from those of traditional microscopes, even scanning probe microscopies.

We cannot think of any microscopy techniques to compare with. Our work does not attempt to compete on image resolution, but explore an entirely different mode of imaging and a new direction for the possible use of Dirac carrier focused beams and “beam lines”.

We acknowledge that is not made sufficiently clear in the manuscript and have added the following sentence(s):

“Third, we consider the question regarding how quantum coherence will influence the operation of the microscope, beyond setting a limit for the image resolution in the tens of nanometer through diffraction effects.” (P26,L11)

And

“While the image resolution of a DFM can never rival those of established electron or scanning probe microscopies, the Dirac fermion microscope could provide new insight in the details of microscopic scattering processes⁸ and interactions with the environment, disorder, adsorbed molecules, quantum dots, which are crucial for sensing, electronics and optoelectronics applications. “ (P27,L11)

Reviewer #2 (Remarks to the Author):

The authors discuss different schemes for realizing an electron microscope based on graphene Dirac electrons.

While the paper surely deserves publication in some form -- after some revision and proofreading, see below -- to my mind it is clearly not suitable for publication in Nat. Commun. The manuscript does not bring up any substantial new idea, it mainly exploits two already recently proposed ideas for creating Dirac electron beams -- Barnard et al. absorptive pinhole collimator (Ref. [45]), Liu et al. parabolic collimator proposal (Ref. [58]) -- to discuss how the resultant beam can be used for microscopy purposes in specific geometries. The collimator studied by the authors, see Figs. 2 - 6, is a simple "series combination" of the pinhole of [45] and the parabola of [58], compare with Fig. 1 from [45] and Fig. 1 from [58]. Though this is evident, it is not very clearly stated, and I find the presentation a bit misleading. Furthermore, the simulations are classical and hence much cruder than those of [58], which appear to be quantum mechanical. The analysis carried out in the experimental work [45] also appears more accurate than the present one, as diffraction effects are explicitly considered in [45].

First, we appreciate the insightful comments and relevant criticism, and we attempt here to clarify our viewpoints and reasons for why we feel that the manuscript deserves consideration to a cross-disciplinary, high impact journal such as Nature Communications.

Combining existing work. The idea to design a 2D electron microscope has been part of the research plan for our center of excellence since May 2016, and the work on the design and simulations started immediately after a conversation between authors PB and CS (Sep 24, 2016). The appearance of two preprints on Arxiv^{9,10}, showed us that two of our essential components, the curved Veselago lens and the aperture, are feasible - and also that top research groups are already moving in this direction. The notion of a parabolic lens as lens element was developed independently by us before we knew of Richter group's preprint. We learned about this on Jan 12, shortly before we submitted the manuscript; this is the main reason we only wrote a short mention and a reference; while Ref¹⁰ is an excellent paper, it had no impact on our manuscript. While we

already used apertures in our microscope design (any ray-based microscope has apertures), the idea of *electrical grounding* of the apertures indeed came from Goldhaber-Gordon group's wonderful preprint⁹, as evident from the manuscript.

In short, had we written that our concept was based on the two preprints (of which one is now published), we would not be telling the truth.

No substantial new idea. The concept of a Dirac fermion microscope has, to our knowledge, not been suggested anywhere in literature. After some searching, we did find a comment posted online by leading Columbia University researcher Cory Dean, where he mentions “on-chip” microscopes and other such possible applications of electron optics (posted Oct 5, 2016, <http://engineering.columbia.edu/news/james-hone-electrons-graphene>). This was on one side encouraging, but naturally also encouraged us to accelerate the work and finish the manuscript.

Indeed, we exploit known physical principles predicted over the past decade and demonstrated mainly over the past 3 years. Proposing a scientific instrument, a microscope, without evidence that the underlying physical principles could work in practice, would be somewhat diffuse and speculative action, which, in our minds, would disqualify the work for high impact publication.

We argue that the concept of a 2D microscope is in fact novel, as are the concrete suggested realisations; we even show how stitching can be implemented to deal with image aberration.

We believe that further development of high-end ballistic devices can be stimulated by having a clear purpose. We are not convinced that there is much more physics to learn by increasing the mean free path from 10 μm to 100 μm , but that would be extremely relevant for a technology point of view – for electron beam devices such as ours. Materials science is driven by the need for better performance.

With this, we hope that more researchers will join the race to utilize electron optics for functional devices, and develop this into a research/engineering field in its own right. Our proposal is intended to contribute to that; to inspire towards developing more advanced and complex composite electron optics devices based on the simple designs we explore.

The following paragraph has been introduced/modified to clarify this:

In the following, however, we exclusively focus on cryogenic temperatures, where both electron-electron and electron-phonon scattering processes are strongly suppressed, and where an upper limit for the mean free path is yet to be determined. With the steadily improving graphene device quality and the numerous confirmations that graphene is capable of supporting transport in the mesoscopic regime ($\ell_{\text{mfp}} > L \gg \lambda_{\text{F}}, \lambda_{\phi}$)^{3,5-7,11-15}, we find that complex instruments that utilize relativistic charge carriers for practical purposes has become realistic. We examine here such an apparatus: a two-dimensional electron microscope, based on Dirac electron optics, which we in the following refer to as a Dirac fermion microscope (DFM). (P3,L8)

Simulations being crude. Reviewer 2 is raising a relevant point. Most papers in electron optics uses quantum calculations in some form to either prove or complement their experimental results. In the mesoscopic limit, which we consider as the natural target for the DFM, the size of the device and the mean free path is much larger than the electron wavelength. Semiclassical (trajectory) calculations have since the late 80's shown to be effective in capturing the overall transport characteristics for mesoscopic devices, using transmission probabilities calculated in classical potentials with semiclassical probability distributions (and scattering cross sections, when needed), to compute multiterminal magnetoconductance, i.e. , based on

seminal works of Buttiker¹⁶ and Houten/Beenakker¹⁷⁻¹⁹, and later for graphene²⁰, a recent paper from Peeters group. We are in no doubt that these are known to the reviewer; we mention this merely to underpin the rationale for our choice of methodology.

Depending on the size and the device geometry, quantum coherence can appear as anything from a perturbation to a dominant effect. By carrying out quantum calculations on micron-size versions of our proposed microscope, we could possibly achieve a higher accuracy in describing behavior, but the relevance of quantum calculations will diminish as the device is scaled up in size. – with the exception of narrow pinholes (as treated in detail in Ref⁹) and the scattering characteristics of very small targets. The scattering of electron waves on a small target depends on the target's shape size, and the exact nature of the interaction with the electron system.

We note that in order to reach micron-length sample sizes in the quantum simulations in Ref⁹ a scaling approximation is used on top of the tight-binding approximation. It is not clear how this approximation works for scattering against various objects in the Dirac microscope, and it also neglects the finite mean-free path; we are not convinced that using this approach would be valid for scattering on small targets. All in all, these calculations, however, demonstrate agreement with the expected classical trajectories and thus substantiate our approach. We are planning to explore what can be expected by imaging different types of targets: plasmonic superstructures, metal dots, quantum dots, defects, and so on.

Our simulations have been carried out in a toy model with a relatively small size of 4 μm , since we wanted to provide numbers and dimensions that are immediately feasible for most groups working with encapsulated graphene. Linear upscaling of all features by a factor of 5-10 should be possible with today's best samples (Stampfer), and there is no reason to believe that this cannot be pushed further.

In summary, we hope to have convinced the reviewer that the ballistic trajectory calculations – although simpler than full quantum calculations – provides an adequate description of mesoscopic Dirac electron optics, if the graphene-specific behavior (Klein-tunneling and Veselago lensing) across pn-junctions is properly modelled. The justification of the methodology relies on quantum transport calculations to converge towards the classical trajectory calculations in the mesoscopic limit, following i.e. Peeters²⁰

We thank the reviewer for pointing us in this direction, and have included the following sentences to make these points clearer:

While size effects such as conductance quantization^{21,22}, Fabry-Perot-like interferences and sidewall roughness in the electron emitter may alter the angular distributions, we consider here the hard mesoscopic limit, where coherence and diffraction effects do not mask the overall behavior²⁰. (P8,L11)

This can, however, be countered by increasing of the aperture size and the carrier density to reduce the Fermi wavelength. Our semiclassical simulations indicate that the image formation is robust towards increasing the aperture size, as we show in Supplementary Fig. 5. As pointed out above, our simulations represent the mesoscopic limit, where electron currents are well approximated by classical trajectories/ray tracing^{12,20}. Upscaling of the lateral dimensions as a route to diminish diffraction and coherence effects will benefit directly from continued development of sample quality^{11,13}. (P26,L15)

REVIEWER #2. Typos and proofreading.

REV2: The beam bending in Fig. 1 (b), (c) is wrong, given the direction of the magnetic field. Checking all other figs. for similar problems is required.

We thank the reviewer for finding this mistake. It has been fixed, and the other figures were checked.

REV2: The colour code of the images is never discussed. Even if arbitrary units are used, I think this should be done to improve clarity. For example in Fig. 3 "bright" should mean high ray (electron current) density, but this is also used for highlighting the targets. Furthermore, certain figure panels seem not to be referred to in the main text, e.g. Fig. 2 d: as far as I understand, in line 23 of p.3 the authors refer to 2 d, but actually mean 2 e. See point 1. about checking for similar mistakes.

Color codes. We do not use/need the information on the quantitative magnitude of the current in the maps (since only the transmission current matters); these serve to illustrate the passage of the electron beam through the device as well as the interaction with target objects. We have introduced the following sentences to account for this,

In caption Figure 3: "Different color scales ranging from 0 (zero current) to 100% (high current) are used to visualize zones (p,n) with different carrier polarity." (P14,L5)

and now also provide color codes. In addition, we also added distinct graphics to more clearly distinguish the targets from the beam currents, as we agree that plotting the targets using the same color scale is misleading. We do however wish to keep the slightly different color scales for the p and n regions, as this makes it much easier to identify areas with same charge carrier polarity, see e.g. Fig. 4a, and hope that the addition of color scales solves the issue satisfactorily.

The mistakes in the Figure panels have been fixed.

REV2: Since graphene is not a topological insulator, what is meant with "topologically protected backscattering" in the introduction?

We agree, and have removed "topologically protected backscattering" from the sentence.

REV2: A scale on figs. such as 3 and 4 would be helpful, e.g. showing explicitly the Fermi wavelength in comparison to the size of the targets.

The calculations represent best the mesoscopic limit, where the wavelength is insignificant²⁰. The wavelength (wavenumber) is needed to calculate the transmission probability and angle across the pn-junctions (using Cayssols approximation formula), however not for the trajectories between the optical elements. We have introduced scale bars in fig. 3 and fig. 4, but also point out the simulations describe the mesoscopic limit:

And have elaborated in the discussion on the role of the finite wavelength and diffraction effects:

Third, we consider the question regarding how quantum coherence will influence the operation of the microscope, beyond setting a limit for the image resolution in the tens of nanometer through diffraction effects. Coherence will for instance influence the angular distribution of electron beams passing through narrow apertures⁹, leading to a broader beam according to the Huygens principle⁷. This can however be countered by increasing of the

aperture size and the carrier density to reduce the Fermi wavelength, pushing the system further into the mesoscopic limit. Our semiclassical simulations indicate that the image formation is robust towards increasing the aperture size, as we show in Supplementary Fig. 5. As pointed out above, our simulations represent the mesoscopic limit, where electron currents is well approximated by classical trajectories/ray tracing^{12,20}. Upscaling of the lateral dimensions as a route to diminish diffraction and coherence effects will benefit directly from continued development of sample quality^{11,13}.(P26,L11)

REV2: Some proofreading would help. A few examples: P. 7, line 19: "singular point"  "single point"; p.26, line 18: "aborptive"  "absorptive"; p.27, line 4 "micron micron"  "micron" .

We have gone through the manuscript carefully, and made these changes as other, including both corrections and improvements of style and clarity.

REVIEWER #3 (Remarks to the Author):

In this manuscript, the authors theoretically design a two-dimensional Dirac fermion microscope in graphene and simulate the detection and the image formation of objects by a beam of ballistic electrons in vacuum chamber. The design and the performance of the key components, including collimators, focusing lenses, deflectors and detectors, are analyzed in detail. The 2D Dirac fermion microscope proposed in this work is scientifically significant and promising for practical applications. However, the authors need to clarify the following issues.

1. We expect that objects to be studied and Dirac fermion microscopes (DFMs) don't necessarily locate in the same graphene platform. Just like a traditional optical/electron microscope, objects can be studied only when they are placed below object lenses. In the current simulations in the manuscript, I understand, the objects studied locate in the same graphene platform as the DFMs. I want to know whether the DFMs can work when objects locate in another graphene platform.

This is an interesting question. Unlike a three-dimensional microscope, the object to be inspected can both be “in-plane”, which in the terms of the reduced dimensionality is “in” the microscope. This would for instance be defects, vacancies or edge roughness. The object to be studied could, however, also be “on-top” of the graphene sheet, i.e. interacting through the hexagonal boron nitride top-layer – i.e. interacting with the 2-dimensional electrons through an interaction which takes place in the third dimension – i.e. a magnetic nanoparticle, which would affect the electron trajectories inside. We think of this as a peculiar, but also exciting feature of the DFM and other such graphene electron optics devices: that it can be accessed from the top, from the third dimension. This corresponds to a vacuum chamber where samples can be put in the sample chamber without opening it. In more practical terms, objects can both be “imaged” either by being located the plane of the graphene, or by being located a distance from it. The other part of the question mentions “another graphene platform”. As we interpret the question, the reviewer envisions that several layers of graphene are involved. We believe that there are such opportunities to make more complex arrangements of graphene structures, i.e. using the techniques for example in Ref²³. We have added the following sentence in the discussion:

Focused beams could possibly also be used to investigate interactions between layers in more complex, multilayered heterostructures²³. (P27, 10)

Is the spatial resolution of the DFMs comparable to traditional electron microscopes?

The electron wavelength in a traditional microscope is typically below nm, and the resolution of SEM can be down to nm-scale. TEM have sub-Ångström resolution. For the DFM the resolution is determined by the electron wavelength as well, which is far larger (in the tens of nm). Achieving similar resolution for graphene DFM as for SEM/TEM is unfortunately impossible. However, it should be noted that the electron wavelength in graphene can be easily tuned over quite some range (e.g. simply by gating the structure), opening up for new opportunities hardly accessible in standard SEM/TEM imaging. In the interest of clarifying this limitation we have added the sentence:

“Third, we consider the question regarding how quantum coherence will influence the operation of the microscope, beyond setting a limit for the image resolution in the tens of nanometer through diffraction effects.” (P26,L11)

REV3: As a collimator, the aperture size in a metal contact aperture connected to electrical ground must be small in order to guarantee the high degree of collimation. Is the ratio of the electrons passing through the aperture to that emitted by the contacts enough high to make sure that the electron beam arriving at objects is strong?

With the wide-open contacts needed for non-diffractive aperture, we are convinced that there will be sufficient signal-to-noise ratio. From what numerous experiments in literature show, even point contacts in the tunnel regime can support sufficient current to do accurate measurements. The signal level will also depend on how well the electron beam current can be collimated and maintained over long distances. It could well be that in early adoptions of large-scale electron optics, signal-to-noise may indeed be an issue, but this is just one of many issues that may arise with complex electron optics devices, and we are not sure what to write about this; time will show. We have added the following sentence:

One potential problem with narrow beams that are guided across long distances by small magnetic fields, is that the sensitivity to weak disorder potentials and variations in the carrier density could lead to signal noise and beam broadening effects. Even with ballistic mean free paths extending beyond the sample size, the beam may still follow irregular paths, as reported in III-V heterostructure 2D electron gas by Jura et al.²⁴. (P25,L20)

REV3: Besides a parabolic lens, an aperture is first used to collimate electron beams. Whether do parabolic lenses work well only when the angular spread of incident electron beams is small? Please clarify this point in the manuscript.

The parabolic lenses will work equally well for narrow and broad angular spread, see Fig. 5, and this was corroborated by the recent paper by Richter's group¹⁰. The use of collimated beams in combination with a parabolic lens is to increase the focal length while keeping the beam diameter small. We have reformulated the caption in Fig 2 to explain this better.

“(f) A combination of an aperture and a parabolic lens produces an electron beam with a long focal length.”(P9,L12)

REV3: Does "hBN" in page 2 refer to hexagonal boron nitride?

We have now written out “hexagonal boron nitride (hBN)”, as should have been done. We thank the reviewer for catching that.

**REV3: Some minor changes must be done in the following sentences:
"where the metal-graphene contact region from the main device area" in page 7.**

This has been corrected.

REV3: "For 10^{12} cm^{-2} " should be explicitly expressed as "For the carrier density $n=10^{12} \text{ cm}^{-2}$ ".

This has been corrected.

REV3: "as illustrated in Fig 2d" should be written as "as illustrated in Fig. 2e" in page 10.

This has been corrected.

REV3: What do "according to (0.1)" refers to in Page 11?

The reference is now changed to “Eq. 1”, and the equation number is changed to “1” as well.

REV3: I can't find Fig. 1f and Fig.3g that are mentioned in page 12 and page 14, respectively.

The references are now changed to Fig 1 and Fig 3d-3e, which should be correct.

REV3: “get a information” in page 15.

The sentence has been reformulated to remove the error.

REV3: (a-d) should be written as (a-c), in the caption part of Fig.4.

The caption has been changed to solve the issue.

REFERENCES

- 1 Katsnelson, M. I., Novoselov, K. S. & Geim, A. K. Chiral tunnelling and the Klein paradox in graphene. *Nature Physics* **2**, 620-625 (2006).
- 2 Cheianov, V. V., Fal'ko, V. & Altshuler, B. L. The focusing of electron flow and a Veselago lens in graphene p-n junctions. *Science* **315**, 1252-1255 (2007).
- 3 Sandner, A. *et al.* Ballistic Transport in Graphene Antidot Lattices. *Nano Letters* **15**, 8402-8406 (2015).
- 4 Lee, M. *et al.* Ballistic miniband conduction in a graphene superlattice. *Science* **353**, 1526-1529 (2016).
- 5 Taychatanapat, T., Watanabe, K., Taniguchi, T. & Jarillo-Herrero, P. Electrically tunable transverse magnetic focusing in graphene. *Nature Physics* **9**, 225-229 (2013).
- 6 Rickhaus, P. *et al.* Snake trajectories in ultraclean graphene p-n junctions. *Nature Communications* **6**, 6470 (2015).
- 7 Lee, G. H., Park, G. H. & Lee, H. J. Observation of negative refraction of Dirac fermions in graphene. *Nature Physics* **11**, 925-929 (2015).
- 8 Guinea, F. Models of Electron Transport in Single Layer Graphene. *Journal of Low Temperature Physics* **153**, 359-373 (2008).
- 9 Barnard, A. W. *et al.* Absorptive pinhole collimators for ballistic Dirac fermions in graphene. *arXiv:1611.05155* (2016).
- 10 Liu, M.-H., Gorini, C. & Richter, K. Creating and Manipulating Electron Beams in Graphene. *arXiv:1608.01730v1* (2016).
- 11 Wang, L. *et al.* One-Dimensional Electrical Contact to a Two-Dimensional Material. *Science* **342**, 614-617 (2013).
- 12 Chen, S. *et al.* Electron optics with p-n junctions in ballistic graphene. *Science* **353**, 1522-1525 (2016).
- 13 Banszerus, L. *et al.* Ballistic Transport Exceeding 28 μm in CVD Grown Graphene. *Nano Letters* **16**, 1387-1391 (2016).
- 14 Bhandari, S. *et al.* Imaging Cyclotron Orbits of Electrons in Graphene. *Nano Letters* **16**, 1690-1694 (2016).
- 15 Taychatanapat, T. *et al.* Conductance oscillations induced by ballistic snake states in a graphene heterojunction. *Nature Communications* **6**, 6093 (2015).
- 16 Buttiker, M. 4-TERMINAL PHASE-COHERENT CONDUCTANCE. *Physical Review Letters* **57**, 1761-1764 (1986).
- 17 Beenakker, C. W. J. & Vanhouten, H. MAGNETOTRANSPORT AND NONADDITIVITY OF POINT-CONTACT RESISTANCES IN SERIES. *Physical Review B* **39**, 10445-10448 (1989).
- 18 Van Houten, H. *et al.* Coherent electron focusing with quantum point contacts in a two-dimensional electron-gas. *Physical Review B* **39**, 8556-8575 (1989).
- 19 Molenkamp, L. W. *et al.* ELECTRON-BEAM COLLIMATION WITH A QUANTUM POINT CONTACT. *Physical Review B* **41**, 1274-1277 (1990).
- 20 Milovanovic, S. P., Masir, M. R. & Peeters, F. M. Spectroscopy of snake states using a graphene Hall bar. *Applied Physics Letters* **103** (2013).
- 21 Terres, B. *et al.* Size quantization of Dirac fermions in graphene constrictions. *Nature Communications* **7** (2016).
- 22 Tombros, N. *et al.* Quantized conductance of a suspended graphene nanoconstriction. *Nature Physics* **7**, 697-700 (2011).
- 23 Gorbachev, R. V. *et al.* Strong Coulomb drag and broken symmetry in double-layer graphene. *Nature Physics* **8**, 896-901 (2012).
- 24 Jura, M. P. *et al.* Unexpected features of branched flow through high-mobility two-dimensional electron gases. *Nature Physics* **3**, 841-845 (2007).

Reviewers' Comments:

Reviewer #2:

Remarks to the Author:

The manuscript underwent marginal revision, and I do not see any reason to change my previous assessment: It is fairly interesting and deserves publication in some form, but is clearly below the standards of Nat. Comms.

Judgments should be made based on the current status of the field: an experimental realization of the pinhole collimator (preprint Barnard et al., Ref. [48] in revised manuscript) and a quantum theoretical treatment of the parabolic lens (Phys. Rev. Lett. Liu et al., Ref.[60] in revised manuscript) are already available, therefore I see no room for the present manuscript in a journal like Nat. Comm.

Reviewer #3:

Remarks to the Author:

The authors have addressed all my issues raised in the previous round. I recommend the paper for publication in Nature Communications.

Reviewer #2: Remarks to the Author:

The manuscript underwent marginal revision, and I do not see any reason to change my previous assessment: It is fairly interesting and deserves publication in some form, but is clearly below the standards of Nat. Comms. Judgments should be made based on the current status of the field: an experimental realization of the pinhole collimator (preprint Barnard et al., Ref. [48] in revised manuscript) and a quantum theoretical treatment of the parabolic lens (Phys. Rev. Lett. Liu et al., Ref.[60] in revised manuscript) are already available, therefore I see no room for the present manuscript in a journal like Nat. Comm.

We are thankful to reviewer #2 for finding the work interesting. As such, we stand by our previous responses regarding the novelty and the choice of simulation method. We do respectfully emphasize that we are not proposing a component, but a system – an apparatus exploiting graphene electron optics for a practical purpose, which we hope can inspire further work. While we consider a rather large Dirac fermion microscope in the mesoscopic range, we find, like reviewer #2, that a weak point of our work could be the semiclassical calculations of scattering on small targets – i.e. objects which are not significantly larger than the electron wavelength. We have therefore carried out atomistic tight-binding calculations in small (100 x 100 nm) regions at a high Fermi energy, to achieve similar relations between target size and Fermi wavelength as considered in the manuscript. We show that this approach provides a reasonably accurate description of the quantum mechanical current density (atomistic bond current) and use this to model three of the scenarios considered in our work: (a) caustic patterns in a circular pn-junction, (b) scattering and current jets as a focused, collimated Dirac fermion beam impinge on a circular pn-junction, and (c) scanning a focused beam using a transverse magnetic field. In the manuscript, we have included a section in the discussion that describes this work:

Upscaling of the lateral dimensions as a route to diminish diffraction and coherence effects will benefit directly from continued development of sample quality^{2,12}. One exception is the scattering of Dirac fermion beams on small objects, where quantum coherence effects may cause significant deviations from the transmission signatures found from our semiclassical trajectory simulations. In Supplementary Note 3, we show maps of the atomistic bond current from atomistic tight binding calculations, for collimated Dirac fermion beams generated by parabolic lenses and narrow apertures. Three cases are presented, to illustrate the main concepts of this work: a caustic pattern corresponding to Fig. 5c, a triangular closed orbit in a circular pn-junction with jet currents corresponding to Fig. 5e, and scanning of a focused beam using the magnetic field across a small circular VD, as in Fig. 4 and Fig. 6. As expected, the simulations show that current density of structures which are large compared to the Fermi wavelength, show reasonable resemblance with the classical calculations, but also that quantum coherence leads to bond current patterns with a richer emission and reflection structure, which may be utilized to extract more detailed information of targets than possible with semiclassical calculations.

In that sense, we anticipate that quantum coherence is indeed an opportunity for developing more advanced functionality of the DFM. With phase-coherent beams, interferometric and even holographic microscopy could give new insight in conductance fluctuations and weak localization, since now individual or sets of trajectories can be selected without need for permanent wires, instead of ensemble averaged.

In supplementary Information, we have introduced a new Supplementary Note 3, and renamed the previous Supplementary Note 3 to 4. We also introduce a new Supplementary Figure 4, and renamed SI Figure 4 and 5, to 5 and 6 respectively. SI Note 3 describes the methodology of the calculations including the scaling approach, as well as the simulation results. The bond current calculations are shown in SI Figure 4.

The calculations show that the semiclassical calculations in the three abovementioned cases (caustics, current jets and bound trajectory of Veselago dot and scanning of collimated beam across a target) capture the essential features of the atomistic calculations, but also that a much richer and more detailed behavior of the transmission images can be expected due to quantum coherence. This is an aspect that we have overlooked, and to our mind is rather exciting. Therefore, we thank the reviewer #2 for urging us to do this work.

We hope that, with these changes and the arguments that we have put forward here and in the previous response, the manuscript can be accepted for publication in Nature Communications.

Reviewer #3:

Remarks to the Author:

The authors have addressed all my issues raised in the previous round. I recommend the paper for publication in Nature Communications.

We thank the reviewer for acknowledging our responses to the raised issue, and for the positive feedback on our work.